# Efficient Recovery of Waste Cotton Fabrics Using Ionic Liquid Methods

**DOI:** 10.3390/polym17070900

**Published:** 2025-03-27

**Authors:** Xiaozheng Zhang, Wenhao Zhou, Wenhao Xing, Yingjun Xu, Gangqiang Zhang

**Affiliations:** College of Textile and Clothing, Institute of Functional Textiles and Advanced Materials, State Key Laboratory of Bio-Fibers and Eco-Textiles, Collaborative Innovation Center of Marine Biomass Fibers Materials and Textiles of Shandong Province, Qingdao University, Qingdao 266071, China; 15550505638@163.com (X.Z.); 15194059841@163.com (W.Z.); xingwenhao0705@163.com (W.X.)

**Keywords:** cellulose, waste cotton fabrics, regenerated cellulose film, ionic liquid

## Abstract

Cotton fiber, renewable natural cellulose, make up the largest portion of textile waste. The ionic liquid method has been successfully employed to regenerate waste colored cotton fabric in this study, offering a comprehensive approach to the recycling of waste cotton. The chemical recovery process for reclaimed cellulose materials is crucial for high-value recycling of waste cotton fabrics. In this study, waste and new, colored and white cotton fabrics were used as experimental subjects. The breaking strength, degree of polymerization, iodine adsorption equilibrium value, and crystallinity between old and new fabrics were investigated. Ionic liquid 1-allyl-3-methylimidazole chloride ([AMIM]Cl) and zinc chloride (ZnCl_2_) were selected to dissolve decolorized waste cotton fabric. Optimal conditions for dissolving the fabric using [AMIM]Cl were investigated. The best dissolution conditions identified were DMSO at a ratio of 1:1 with a dissolution temperature of 110 °C over a duration of 120 min. Additionally, the optimal film formation parameters included a solution concentration of 6%, solidification time of 3 min, and solidification bath temperature of 0 °C. Regenerated cellulose films from both the ionic liquid system (A-film) and zinc chloride system (Z-film) were prepared. The characteristics of the film produced using the most advanced technology were systematically investigated and evaluated. The results of this study provide a crucial theoretical foundation for the recovery and regeneration of waste cotton fabrics.

## 1. Introduction

In recent years, there has been a global emphasis on environmental issues concerning the sustainable utilization of natural resources. Since 2017, the annual extraction and consumption of primary resources worldwide have surpassed 100 billion tons, while the annual production of municipal solid waste exceeds 2 billion tons—a figure projected to increase by 70% by 2050 [1,2]. Among these, waste textiles represent a significant portion of municipal solid waste. If a substantial amount of waste textiles is not recycled, it can lead to environmental pollution and squander renewable resources [3]. Many developed countries such as the United States, Germany, Japan, and Sweden have established relatively comprehensive systems for recycling waste textiles that align with green development principles. In contrast, China’s system for recycling waste textiles remains underdeveloped, with a recycling rate estimated at only about 20%. Waste textiles contain harmful substances such as dyes and finishing agents that can result in severe soil and air pollution when disposed of in landfills or incinerated. Cotton fabric is particularly noteworthy due to its high cellulose content; it constitutes the largest volume of renewable resources globally and represents the predominant category within textile materials [4,5]. The recycling of waste cotton textiles holds significant potential for mitigating environmental pollution while enhancing resource utilization rates [6,7]. Thus, promoting the recycling of waste cotton fabric is crucial for conserving resources, reducing pollution emissions and carbon footprints. It plays an essential role in improving resource efficiency, decreasing industrial energy consumption, and establishing a green low-carbon circular economy.

There are three primary methods for recycling waste cotton fabric: physical methods, chemical methods, and energy recovery methods. The physical method involves altering the physical form of waste textiles through processes such as shearing, press grinding, spinning, or physical mixing. In this approach, the chemical structure of the constituent materials in the waste textiles remains intact [8]. In contrast, the chemical method entails dissolving or modifying waste textiles using chemical means to reduce their degree of polymerization or molecular weight. This process ultimately allows for the preparation of recycled cellulose or cellulose derivatives via polymerization or spinning. The energy recovery method focuses on generating energy through incineration or chemical conversion for power generation purposes [9,10]. However, products derived from physical recovery tend to be of lower quality and have a reduced utilization rate [7,11]. Additionally, waste textiles often contain substances like dyes and finishing agents that can release toxic gases during incineration, leading to significant air pollution. While chemical recovery can yield recycled cellulose and facilitate effective recycling efforts, both physical and chemical recovery present challenges in achieving optimal results [12]. Therefore, employing chemical dissolution techniques on waste textiles to produce recycled cellulose holds considerable promise for future applications.

Due to the strong hydrogen bonding between cellulose molecules, along with its high degree of orientation and crystallinity, cellulose is challenging to dissolve in common solvents [13,14]. This difficulty hinders the development of cellulose resource reuse. Currently, most commercial processes utilize the viscose method for dissolving cellulose; however, this process significantly contributes to environmental pollution. In light of rapid economic development, there is an increasing emphasis on ecological preservation by governments. Ionic liquids are often referred to as “solvents of the future” due to their low vapor pressure, resistance to volatilization, ease of recovery, and lack of harmful gas emissions during industrial use [15,16]. Compared with traditional solvents, ionic liquids offer unique advantages in terms of thermal stability and electrical conductivity. Ionic liquids consist entirely of cations and anions. In 2003, Ren Qiang et al. [17] synthesized 1-allyl-3-methylimidazole chloride ([AMIM]Cl) using N-methylimidazole, allyl chloride, and chlorobactene as raw materials; they discovered that it exhibited good solubility for cellulose. Kallidanthiyil et al. [18] prepared alkaline ionic liquids (ILs) with unconventional organic anions and used it for cellulose dissolution studies. Asaadi et al. [19] employed a type of ionic liquid known as [DBNH]OAc to dissolve waste cotton fabric and produce regenerated cellulose fibers. Lv et al. [20] demonstrated that [AMIM]Cl could effectively separate cotton fibers from waste blended fabrics through a straightforward process while yielding regenerated cellulose with excellent thermal stability. Zheng et al. [21] employed [AMIM]Cl to effectively dissolve cellulose and subsequently produce high-performance regenerated cellulose films. The aqueous solution of ZnCl_2_ was first reported as a solvent for cellulose by Letters et al. [22] in 1932. ZnCl_2_ is an environmentally friendly inorganic metal salt characterized by minimal environmental toxicity or volatility [23]. Ma et al. [24] demonstrated that the hydrated zinc ions in ZnCl_2_ solutions exhibit ideal characteristics of migration, penetration, interaction, and dispersion, thereby effectively dissolving cellulose at room temperature. Most importantly, this type of inorganic metal salt can be easily recycled.

As a composite material derived from cellulose, research and development into regenerated cellulose films hold significant practical importance for renewable resource utilization, ecological improvement efforts, and sustainable economic growth.

This paper focuses on dyed textile waste, comparing waste white cotton, new white cotton, waste dyed cotton, and new dyed cotton. The strength, degree of polymerization, iodine adsorption balance, and crystallinity were examined. Ionic liquid [AMIM]Cl and ZnCl_2_ aqueous solutions were employed to effectively dissolve and decolorize the cotton fabrics. The dissolution process of [AMIM]Cl was discussed, along with the solubility and film-forming properties of decolorized cotton fibers in both dissolution systems. Additionally, the spinning capabilities of the cellulose solutions prepared from these systems were explored, providing a theoretical foundation for recycling waste colored cotton fabrics.

## 2. Materials and Methods

### 2.1. Experimental Materials

Experimental materials: As shown in Figure 1, from (left) to (right), waste white cotton fabric (A); new white cotton fabric (B); waste dyed cotton fabric (C); new dyed cotton fabric (D). Copper ethylenediamine solution (AR); Sodium hydroxide (AR); 0.1mol/L Iodine standard solution (AR); Soluble starch (AR); Sodium thiosulfate pentahydrate (AR); Anhydrous sodium sulfate (AR); Urea (AR); N, n-dimethylacetamide (AR); Glycerol (AR); Zinc chloride (AR); 1-allyl-3-methylimidazole chloride (AR); and Dim ethylidene Sulfone (DMSO) (AR).

### 2.2. Experimental Methods

#### 2.2.1. Determination of Polymerization Degree of Cotton Fabrics

According to GB/T 1548-2016 [25] “Determination of pulp viscosity” (ISO 5351:2010) [26], the degree of aggregation of cellulose was determined by using a 1835 type Ubbelohde Viscometer (capillary diameter of (0.84 ± 0.05) mm).$$\begin{array}{c} \left[\eta \right]=\frac{\eta_{sp}}{C}=\frac{\eta_{sp}}{0.05}\left(dL/g\right)=\frac{t_{1}-t_{0}}{0.05t_{0}}\;\left(dL/g\right)\\ \bar{DP}=\frac{\left[\eta \right]}{K}=\frac{\left[\eta \right]}{8\times 10^{-3}\left(dL/g\right)}=125\left(g/dL\right)\left[\eta \right] \end{array}$$

*DP*—Degree of polymerization, the number of repeating units in the polymer chain.

*C*—Concentration of the standard copper ethylenediamine solution (g/dL).

*K*—Consistency index, a parameter used to describe the relationship between fluid viscosity and temperature change (dL/g).

[*η*]—Intrinsic viscosity, reaction of internal friction between polymer and solvent molecules (dL/g).

η*_sp_*—Increased viscosity, internal friction between polymers, and between pure solvents and polymers.

*t*_1_—the time it takes for the sample solution to flow through the Ubbelohde viscometer (S).

*t*_0_—the time it takes for the standard copper ethylenediamine solution to flow through the Ubbelohde viscometer (S).

#### 2.2.2. Determination of Accessibility of Cotton Fabrics (Iodine Equilibrium Adsorption Value)

The crushed fabrics of groups A, B, C, and D were dried in the oven for 6 h, weighed with an analytical balance about 60 mg for each tissue, and put into a brown glass bottle with a ground seal. Then, we accurately added 5 mL iodine solution (0.100 mol/L), accurately added 5 mL saturated Na_2_SO_4_ solution, shook the brown glass bottle so that the cotton fabric in the bottle and the iodine solution are evenly mixed, and placed in a 20 °C constant temperature water bath. We shook the bottle frequently, removed the brown glass bottle from the water bath for 1 h, and filtered the cotton fiber. The standard sodium thiosulfate solution (0.1 mol/L) was titrated with 1% starch solution as an indicator, and the blank test was carried out with the same operation. Iodine equilibrium adsorption value (V_IS_) is$$V_{IS}=C\times \frac{\left(a-b\right)\times c}{W}=Iodine\;adsorption\;in\;milligrams/g$$

C—Relative atomic mass of iodine atom = 127 g/mol.

a—Na_2_S_2_O_3_ solution consumed in blank test (mL).

b—Sample consumed Na_2_S_2_O_3_ solution (mL).

c—Molar concentration of Na_2_S_2_O_3_ solution (mol/L).

W—Dry mass of the sample (g).

#### 2.2.3. Instrument Testing

The specific surface area of four groups of crushed cotton fabric was determined by Micromeritics surface aperture tester. The measurement was carried out according to the test method in the national standard [27].

The microscopic dissolution state of waste cotton fabric was observed and photographed with an S650 microscope.

According to the ISO standard method, the mechanical properties of the film at room temperature were measured by using the American Instron 3300 universal material testing machine (Instron, Norwood, MA, USA).

The viscosity of cellulose solution was measured at 5–200 rpm at room temperature by DV-1 Prime digital rotary viscometer (AMETEK Brookfield, Middleborough, MA, USA).

The UV2700 UV-VIS spectrophotometer (Shimadzu, Kyoto, Japan) was used to test the transparency of different films at 350–750 nm wavelengths, using air as a reference.

The infrared absorption spectra of decolorized fabrics and regenerated cellulose films were measured by Nicolet iS50 infrared spectrometer (Thermo Fisher Scientific, Waltham, MA, USA). The scanning range is 400–4000 cm^−1^, with 32 scans.

The crystal structure spectra of the regenerated cellulose films were determined using a D8 Advance X-ray diffractometer (Bruker, Billerica, MA, USA), which obtained 3000 counts in the range of 10–60° at a scanning speed of 2°/min and a step size of 0.01°.

The element composition of the regenerated cellulose film was determined by KRATOS AXIS-ULTRA DLD spectrometer (Kratos Analytical Ltd., Kyoto, Japan).

A Tuscan Vega3-SBU (Tescan, Brno, Czech Republic) was used to study the element distribution of surface morphology of regenerated cellulose films at 20.00 kV. A gold sample was sprayed for 1 min before the test.

### 2.3. Dissolution of Waste Cotton Fabric

#### 2.3.1. Zinc Chloride System

The pulverized waste cotton fabric was placed in a 250 mL flask. Subsequently, the ZnCl_2_ powder was rapidly transferred into the flask, followed by the addition of the required amount of water to produce zinc chloride solutions with varying concentrations (70%). The flask was then positioned on a constant temperature magnetic stirrer set at 80 °C, equipped with a Teflon stirring rod and a stopper. A mechanical stirrer was employed to vigorously mix the solution at a speed of 600 rpm, resulting in cellulose solutions of different concentrations (5 wt%). During the experiment, samples were taken using a glass rod for observation under an optical microscope to monitor and document the dissolution process.

#### 2.3.2. Ionic Liquid System

The completely crushed waste cotton fabric was placed into a fully dried three-way flask. Under a nitrogen atmosphere, ionic liquid and DMSO were added to the flask in a weight ratio of (0.5:1–2.5:1). The oil bath was then heated to a specific temperature ranging from 80 to 120 °C, while the mixture was thoroughly stirred using a high-speed mixer. Cellulose solutions with varying concentrations (2–7 wt%) were obtained as a result. During the experiment, samples were taken with a glass rod for observation under an optical microscope to monitor and document the dissolution process.

### 2.4. Preparation of the Regenerated Cellulose Film

We took the filtered cellulose solution and doubled it in a vacuum oven. Subsequently, we spread it onto a glass plate, scraped off the film, and immersed it in distilled water at varying temperatures (0–40 °C) for solvent–non-solvent ion exchange solidification. A glycerol solution (1–15%) was utilized in the solidification bath of the zinc chloride system. We allowed sufficient time for the solvent to dissolve within the solidification bath (3–30 min). Afterward, we removed the film and placed it on a glass surface dish. The A-film and Z-film were obtained by drying at room temperature. The preparation process of the regenerated cellulose film is illustrated in Figure 2.

## 3. Results

### 3.1. Comparison of Properties Between Old and New Cotton

#### 3.1.1. Analysis of Breaking Strength and Elongation at Break of Old and New Cotton Fabrics

As shown in Figure 3, the average breaking strength of waste white cotton fabric is (132.30 ± 0.20) N, with an average breaking elongation of (55.97 ± 0.57)%. In contrast, the new white cotton fabric exhibits a higher average breaking strength of (234.60 ± 0.36) N, an average breaking elongation of (54.37 ± 0.29)%. For waste dyed cotton fabric, the average breaking strength is recorded at (211.93 ± 0.12) N, accompanied by an average breaking elongation of (52.44 ± 0.52)%. Conversely, the newly dyed cotton fabric demonstrates a superior performance with an average breaking strength of (236.10 ± 0.11) N, an impressive average breaking elongation of (59.54 ± 0.33)%. According to these test results, it is evident that both the breaking strength and elongation properties have diminished in comparison to those observed in new pure cotton fabrics for waste pure cotton fabrics to varying extents [28]. This phenomenon may be attributed to several factors including exposure to washing processes, sunlight degradation, perspiration effects, and oxidative reactions from atmospheric oxygen during regular use—each contributing to a reduction in fiber strength over time [29]. Additionally, it is plausible that a decrease in the degree of polymerization within the fibers has led to diminished values for both fiber-breaking strength and elongation at break.

The breaking strength of fabrics is significantly influenced by parameters such as cotton variety, yarn count, and fabric structure. In this study, all fabrics were sourced from online platforms, resulting in a considerable degree of randomness that makes it challenging to control these parameters. Among these factors, the cotton variety can be identified; specifically, the fabric used is consistent with fine cashmere cotton. Furthermore, all fabrics employed in this experiment are thin woven types, exhibiting minimal variation in structural parameters.

It is important to note that the yarn count for each fabric remains unspecified. A higher yarn count indicates greater linear density and enhanced breaking strength. Consequently, the breaking strength data presented in this study serve only as a preliminary comparison between new and old cotton varieties. For more detailed analysis, it would be necessary to further reduce variables and conduct more rigorous experimental research.

#### 3.1.2. Analysis of Polymerization Degree of Old and New Cotton Fabrics

The polymerization degree of fibers in four different tissues was assessed using the viscosity method. As shown in Figure 4, the average polymerization degree of waste white cotton fabric was found to be 881 ± 17, while that of new white cotton fabric measured at 1355 ± 40. In contrast, the average polymerization degree for waste dyed cotton fabric was recorded as 465 ± 15, and the newly dyed cotton fabric exhibited an average polymerization degree of 1848 ± 21. The results indicate that, when compared to new pure cotton fabric, there is a notable decline in the polymerization degree of waste pure cotton fabric. Furthermore, the polymerization degree of newly dyed pure cotton fabric is significantly higher than that observed in other fabrics. This discrepancy may be attributed to a higher concentration of dyes and finishing agents present in the newly dyed fabrics, which could enhance the viscosity of cellulose’s copper ethylenediamine solution [30,31]. Additionally, it is possible that this solution’s viscosity exceeds the optimal range suitable for measurement by Ubbelohde viscometer.

#### 3.1.3. Analysis of Accessibility of Old and New Cotton Fabrics

The iodine equilibrium adsorption value (V_IS_) serves as an indicator of cellulose availability. The adsorption of iodine by cellulose occurs on the surfaces of both the microcrystalline and non-crystalline zones, while it does not penetrate into the crystalline zone. Therefore, changes in the accessibility of cellulose within textiles can be assessed by measuring the iodine equilibrium adsorption values of new and used pure cotton fabrics. As shown in Figure 5, the V_IS_ for waste white cotton fabric was determined to be 402 ± 10. In comparison, new white cotton fabric exhibited a V_IS_ of 413 ± 12; waste dyed cotton fabric had a V_IS_ of 465 ± 14; and new dyed cotton fabric recorded a V_IS_ of 508 ± 13. These results indicate that the accessibility of white fabrics is lower than that of dyed fabrics, with new fabrics demonstrating slightly higher accessibility compared to their used counterparts [32].

#### 3.1.4. Analysis of Crystallinity of Old and New Cotton Fabrics

As illustrated in Figure 6, the X-ray diffraction (XRD) patterns of waste white cotton textiles, waste dyed cotton textiles, new white cotton textiles, and new dyed cotton textiles were analyzed to investigate the structure and diffraction peak intensity of four groups of cellulose crystals. All sample exhibits four distinct peaks at 2θ = 15.1°, 16.7°, 22.9°, and 34.8° [20,33], corresponding to diffraction peaks located on the (1–10), (110), (200), and (004) crystal planes. The results indicated that there was no significant difference in the crystal structure of cellulose among these four groups; all samples exhibited cellulose type I [34,35]. This finding suggests that the crystal structure of cellulose remains unaffected by the use of different types of cotton fabrics. From the analysis of diffraction peak intensity, it is evident that undyed cotton fabric demonstrates a significantly stronger peak compared to dyed counterparts. The diffraction peak strength for white cotton textiles shows no notable variation; however, the strength for waste dyed cotton textiles is higher than that observed in new dyed cotton textiles but lower than both types of white cotton textiles. These observations imply that crystallinity among the four groups of cellulose has minimal correlation with their respective strengths.

#### 3.1.5. Analysis of Specific Surface Area of Old and New Cotton Fabrics

The BET test results are presented in Figure 7. The specific surface area of waste white cotton fabric is measured at 1.3836 m^2^/g, with an average pore size of 4.4269 nm. In contrast, the new white cotton fabric exhibits a specific surface area of 1.5251 m^2^/g and an average pore diameter of 4.4374 nm. For the waste dyed pure cotton fabric, the specific surface area is recorded as 1.4792 m^2^/g, accompanied by an average pore size of 4.6590 nm. Conversely, the newly dyed cotton fabric shows a lower specific surface area of 0.8548 m^2^/g and an average pore diameter of 4.4710 nm. These findings indicate that the specific surface area of the new white fabric is slightly greater than that of its older counterpart; however, the newly dyed fabric demonstrates a reduced specific surface area compared to the old dyed variant, while there is minimal variation in average pore diameters across all four types examined [36].

### 3.2. Process Investigation

#### 3.2.1. Dissolving Process

During the dissolution process, no harmful gases are produced, and the ionic liquid exhibits excellent thermal stability. The process operates at low temperatures without requiring pressure, minimizing environmental impact and reducing energy consumption during production [37].

The cotton fabric, after undergoing decolorization, was dissolved in ionic liquid, with the corresponding results illustrated in Figure 8. As depicted in Figure 8a, allyl imidazole ionic liquids serve as effective solvents for cotton fabrics. When the dissolution temperature is increased from 80 °C to 120 °C, the solubility rises from 6.4% to 8.8%. In Figure 8b, at a waste cotton content of 2 wt%, the dissolution time of [AMIM]Cl for cellulose is recorded at 36 min; however, when the solubility increases to 6 wt%, this dissolution time nearly triples. These findings indicate that the presence of cellulose significantly influences the dissolution rate of ionic liquid, with an increase in cellulose content leading to prolonged dissolution times.

As shown in Figure 8c, elevating the dissolution temperature accelerates the breaking of hydrogen bonds between cellulose and allyl groups; consequently, ionic liquid continues to penetrate and dissolve cellulose more effectively while reducing overall dissolution time [38]. Therefore, a temperature of 110 °C was selected as optimal for ionic liquid dissolution. During this process, it was observed that viscosity of the dissolved solution escalated with increasing amounts of cotton fabric present. Once cellulose content reaches a certain threshold, fluidity diminishes significantly due to pronounced pole climbing phenomena—this complicates mechanical agitation and impedes further cellulose dissolution within the ionic liquid. To mitigate solution viscosity and enhance dissolving efficacy, organic solvent DMSO was mixed with [AMIM]Cl [39]. Given that ionic liquids are entirely composed of cationic ions and exhibit minimal reactivity with other substances while maintaining good electrical conductivity, ultimately, [AMIM]Cl: DMSO = 1:1 emerged as an ideal condition for successful dissolution.

The dissolution temperature and dissolution time significantly influence the dissolution efficacy, with optimal conditions identified as 110 °C and a duration of 120 min. The concentration of the solution, solidification time, and solidification bath temperature markedly affect the properties of the regenerated cellulose film. Optimal conditions are established at a concentration of 6%, a solidification time of 3 min, and a solidification bath temperature of 0 °C.

#### 3.2.2. Regeneration Process

(1) Effect of Solidification Time on the Mechanical Properties and Permeability of the Regenerated Cellulose Film

As illustrated in Figure 9, a solution was prepared by dissolving 6% waste cotton fabric in deionized water at 0 °C to investigate the impact of solidification time on the permeability and mechanical properties of the regenerated cellulose film. As the solidification time increased from 3 to 30 min, the elongation at break decreased from 7.4% to 6.8%. Concurrently, the fracture strength of the film diminished from 134 MPa to 86 MPa. The transmittance of the regenerated cellulose film exhibited an initial increase followed by a subsequent decrease. These findings indicate that both fracture strength and permeability are linearly related to solidification time, with an optimal solidification duration identified as being three minutes for this study.

(2) Influence of Coagulation Bath Temperature on Mechanical Properties and Permeability of the Regenerated Cellulose Film

As illustrated in Figure 10, a 6% solution of waste cotton fabric was prepared, with a solidification time of 3 min. This study investigates the effect of coagulation bath temperature on the permeability and mechanical properties of the regenerated cellulose film. It was observed that as the temperature of the coagulation bath increased, the permeability of the film gradually decreased. The fracture strength diminished from 132 MPa to 97 MPa, while elongation at break reduced from 7.3% to 6.9%. This decline may be attributed to elevated temperatures leading to impaired ionic liquid diffusion [40], which facilitates the formation of gel particles and adversely affects the uniformity of the film. Consequently, this results in a significant reduction in permeability, further impacting its mechanical properties. Therefore, it is recommended to set the coagulation bath temperature at 0 °C.

(3) The Effect of Film-Making Solution Concentration on the Mechanical Properties and Transmittance of the Regenerated Cellulose Film

As illustrated in Figure 11, this study investigates the influence of film-making solution concentration on the permeability and mechanical properties of the regenerated cellulose film under deionized water at 0 °C, with a solidification time of 3 min. As the amount of waste cotton fabric increases, the transmittance of the film initially rises before subsequently declining, peaking at a concentration of 6%. The elongation at break for the films increased from 6.7% to 7.4%, followed by a decrease to 5.0%. Additionally, the fracture strength exhibited an increase from 85 MPa to 138 MPa before dropping back down to 95 MPa, demonstrating a trend characterized by initial growth followed by reduction. After comprehensive evaluation, a concentration of 6% for the film-making solution was selected for subsequent experiments.

#### 3.2.3. Solubility of Waste Cotton Fabric

In order to investigate the dissolution process of waste cotton fabric in an ionic liquid solution, waste cotton fabric was introduced into a mixed solution of [AMIM]Cl and DMSO at a 1:1 ratio, maintained at 110 °C. Microscopic images were captured at various dissolution times. As illustrated in Figure 12, the fibers began to swell after 5 min of dissolution; Figure 12b indicates that a significant quantity of fibers disappeared after 15 min. In Figure 12c, the fibers are completely dissolved, leaving only a few fragmented remnants. Figure 12d shows that these broken fibers gradually diminish after one hour of dissolution, indicating that the process is approaching its final stage. By Figure 12e, no insoluble matter remains in the dissolved solution aside from minimal air bubbles. This demonstrates that waste cotton can be effectively and rapidly dissolved within this system.

The underlying mechanism for this dissolution is as follows: the allyl group present in [AMIM]Cl forms hydrogen bonds with hydroxyl groups on cellulose molecules, resulting in enhanced solubility for cellulose [41,42]. Figure 12f serves as a physical reference image depicting the cellulose solution following two hours of dissolution. It is evident from Figure 12f that the resultant solution is clear and transparent, facilitating subsequent experimental procedures.

#### 3.2.4. Rheological Properties of Waste Cotton Fabric

The following figure examines the impact of varying concentrations of waste cotton fabric on the viscosity of ionic liquid solutions at room temperature (25 ± 2 °C). As illustrated in Figure 13, an increase in the content of waste cotton fabric leads to a disruption and reorientation of cellulose molecular chains, resulting in reduced friction between these chains. Consequently, the viscosity of the solution generally decreases. With prolonged shear time, the orientation of cellulose molecules reaches its limit [43,44], causing a gradual and slow decline in viscosity. Notably, when the concentration of waste cotton solution rises to 7%, there is a significant increase in viscosity as depicted in the figure; further increases in solution concentration may lead to gelation.

### 3.3. Representation

#### 3.3.1. FT-IR Analysis

The functional groups of waste cotton fabric and the regenerated cellulose film after decolorization were analyzed using FT-IR spectroscopy, with the results presented in Figure 14. The spectrum of d-cotton closely resembles that of the A-film and Z-film, indicating that no derivative reactions occurred during the dissolution and regeneration processes, thus confirming that the cellulose remained unmodified. The absorption peak at 3400 cm^−1^ corresponds to the stretching vibration of the –OH bond within cellulose molecules, while the absorption peak at 2920 cm^−1^ is attributed to the stretching vibration of –CH bonds, both characteristic features of cellulose [45,46].

In comparison to d-cotton, a new absorption peak was observed at 890 cm^−1^ in the regenerated cellulose film. This peak arises from the deformation vibrations of CH_2_ and C–O–H groups, suggesting that cellulose present in cotton fibers underwent dissolution and regeneration. Furthermore, this transformation indicates a change in crystalline structure from cellulose I to cellulose II [47,48].

#### 3.3.2. XPS Analysis

To investigate the microstructure of regenerated cellulose films, we analyzed the surface elements of films prepared using both zinc chloride and ionic liquid systems through X-ray photoelectron spectroscopy (XPS). As illustrated in Figure 15, the low-resolution XPS spectrum of the A-film reveals distinct characteristic peaks at 293.5 eV (C1s) and 529.7 eV (O1s). Furthermore, in comparison to the A-film, the Z-film exhibits characteristic peaks corresponding to zinc and chlorine elements at 1045.9 eV (Zn 2p) and 199.7 eV (Cl 2p), whereas no residual chlorine is detected in the A-film [49,50]. In summary, as demonstrated by the data presented above, the membrane produced using the ionic liquid system contains no residual chlorine. In contrast, the corresponding element is present in the membrane fabricated through the ZnCl2 system. Therefore, it can be concluded that the ionic liquid system offers greater advantages in terms of environmental protection and sustainability.

#### 3.3.3. XRD Analysis

The results of X-ray diffraction demonstrate the effects of decolorization, dissolution, and regeneration on the crystallinity of cellulose. Figure 16 presents the XRD patterns for d-cotton, the Z-film, and A-film. The d-cotton sample exhibits four distinct peaks at 2θ = 15.1°, 16.7°, 22.9°, and 34.8° [20,33], corresponding to diffraction peaks located on the (1–10), (110), (200), and (004) crystal planes. Following dissolution and regeneration processes, both the Z-film and A-film display a broad peak in the range of 2θ = 15–25°. This encompasses an amorphous cellulose peak at approximately 14.76° as well as overlapping peaks associated with two cellulose II phases at 21.1° (110) and 22.65° (200) [51]. These findings indicate that the fiber crystal structure of cotton transitions from cellulose type I to cellulose type II after undergoing dissolution and regeneration via ionic liquid treatment.

According to the deconvolution analysis, the total crystallinity values for the Z-film (29.2%) and the A-film (21.3%) are significantly lower than that of d-cotton (66.1%). This reduction can be attributed to the disruption and subsequent reformation of hydrogen bonds within cellulose during the dissolution–regeneration process applied to waste cotton fabric, leading to a substantial decrease in both crystallinity and orientation of cellulose fibers. Furthermore, it is noteworthy that the intensity of diffraction peaks for the Z-film is slightly higher than that observed for the A-film; this discrepancy may be due to residual metal zinc present in the regenerated cellulose film.

#### 3.3.4. SEM Analysis

As illustrated in Figure 17, the SEM analysis reveals that the A-film exhibits a smooth surface and a uniform, dense cross-section. This observation suggests that the film solidified at a moderate rate during the forming process, resulting in a defect-free epidermal layer [52]. In contrast to the A-film, the cross-sectional structure of the Z-film is not uniform; phase separation occurs, leading to slight compaction issues and incompleteness. These findings are largely consistent with the results obtained from mechanical properties testing.

## 4. Conclusions

In this paper, new white cotton fabric, waste white cotton fabric, new colored cotton fabric, waste colored cotton fabric as experimental objects, the influence of use and color on the breaking strength, polymerization degree and accessibility of cotton fabric were investigated. Subsequently, the waste colored cotton fabric was dissolved using an ionic liquid and zinc chloride system. The process of dissolving waste cotton through the ionic liquid system was explored further, leading to the preparation of regenerated cellulose membranes. The characteristics of the Z-film and the A-film were analyzed using X-ray diffraction (XRD), X-ray photoelectron spectroscopy (XPS), and scanning electron microscopy (SEM). The conclusions were drawn as following:The iodine equilibrium adsorption value of the new pure cotton fabric is slightly higher than that of the waste fabric, indicating enhanced accessibility. However, the crystallinity of the waste fabric remains unchanged and even exhibits a higher XRD intensity compared to that of the new dyed fabric. Notably, there is no direct linear relationship between cellulose crystallinity and fabric strength. BET test results reveal no correlation between the specific surface area of long-term used fabrics and that of newly produced fabrics.In an ionic liquid system with a composition ratio of [AMIM]Cl: DMSO = 1:1, optimal dissolution conditions are achieved at 110 °C for 120 min; ideal forming conditions include a film solution concentration of 6%, solidification time of 3 min, and a solidification bath temperature maintained at 0 °C.Under optimal processing conditions, the Z-film demonstrates lower mechanical properties (91.0 MPa) and transparency (89.3%) in comparison to the A-film (139.6 MPa) and (93.0%). Surface morphology analysis indicates that the Z-film possesses a defective epidermal layer, whereas the A-film features a smooth surface with uniform density throughout its cross-section. Deformation vibration absorption peaks for CH_2_ and C–O–H appear at 890 cm^−1^ for both films; however, the Z-film shows slightly higher crystallinity (29.2%) than the A-film (21.3%).

The mechanical properties (such as breaking strength and elongation at break) and polymerization degree of waste cotton fabric are generally lower than those of new cotton fabric. This indicates that the fiber properties have deteriorated during use. The accessibility of dyed cotton fabrics is greater than that of white cotton fabrics, suggesting that the dyeing process may enhance fiber accessibility. The crystallinity of the fibers was significantly reduced during dissolution and regeneration, indicating substantial changes in the crystal structure of the fibers throughout these processes. The regenerated cellulose membrane produced using an ionic liquid system exhibits superior mechanical properties and light transmittance compared to that prepared with a zinc chloride system. These results from statistical analysis provide a crucial theoretical foundation for the recovery and regeneration of waste cotton fabrics.

## Figures and Tables

**Figure 1 polymers-17-00900-f001:**
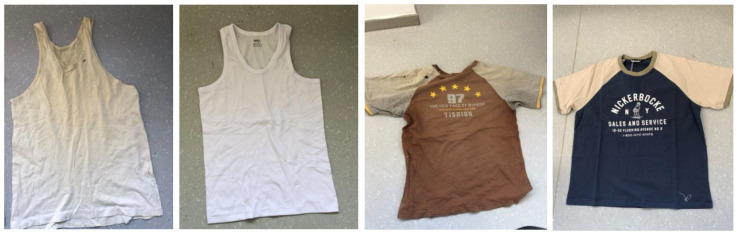
From (**left**) to (**right**), waste white cotton fabric, new white cotton fabric, waste dyed cotton fabric, new dyed cotton fabric.

**Figure 2 polymers-17-00900-f002:**
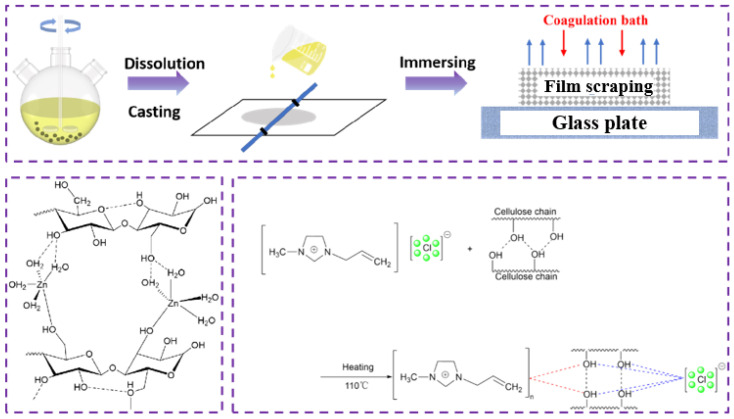
Preparation process and reaction mechanism of the regenerated cellulose film.

**Figure 3 polymers-17-00900-f003:**
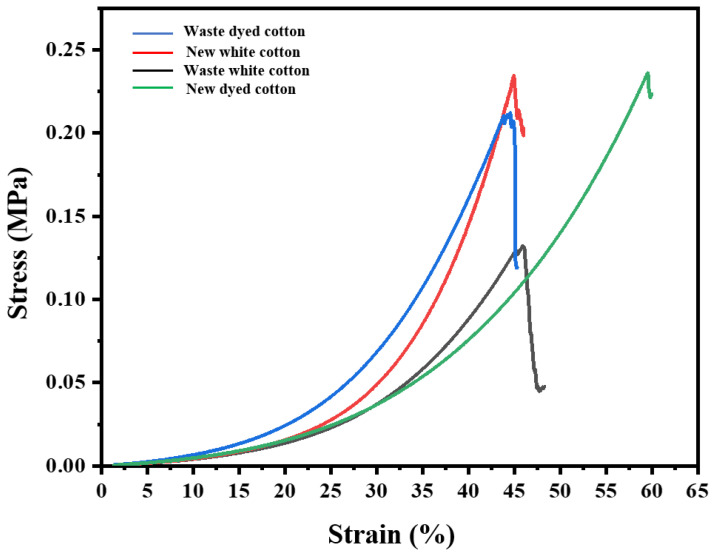
Relationship between stress and tensile strain of waste white cotton fabric.

**Figure 4 polymers-17-00900-f004:**
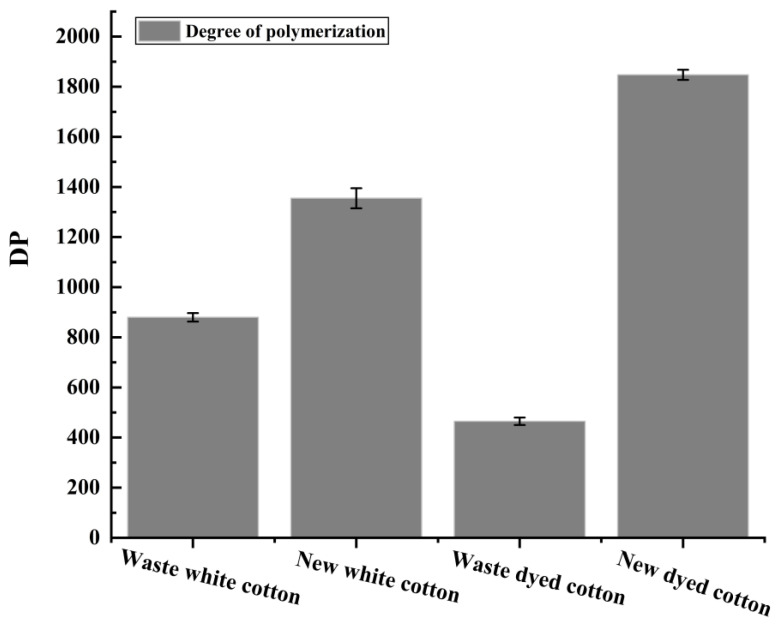
The degree of polymerization of four groups of textiles.

**Figure 5 polymers-17-00900-f005:**
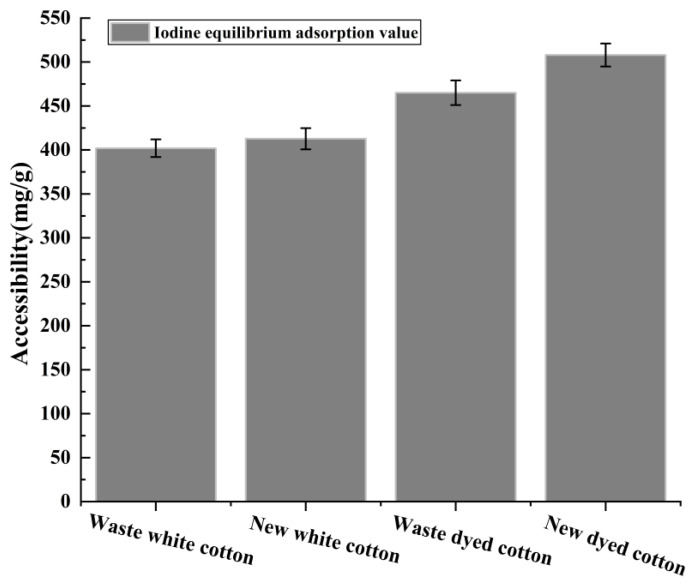
The accessibility of four groups of textiles.

**Figure 6 polymers-17-00900-f006:**
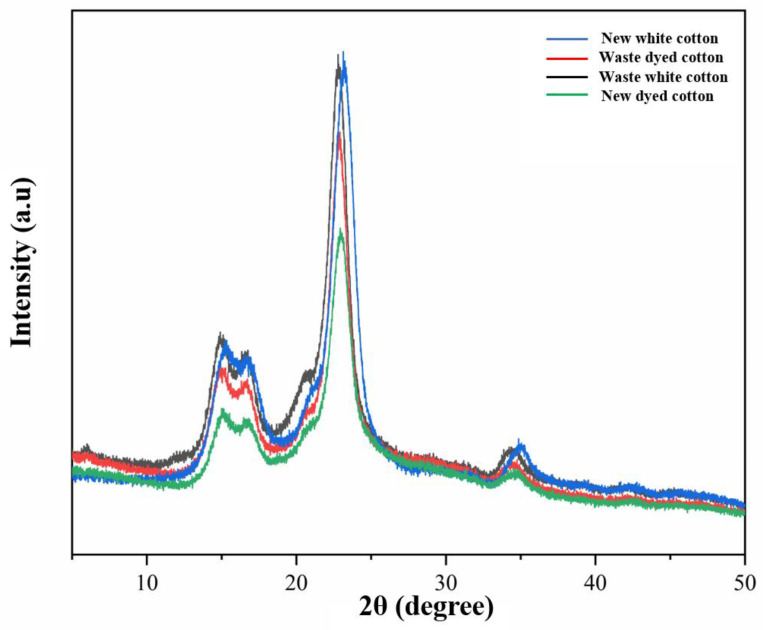
X-ray diffraction spectra of four groups of samples.

**Figure 7 polymers-17-00900-f007:**
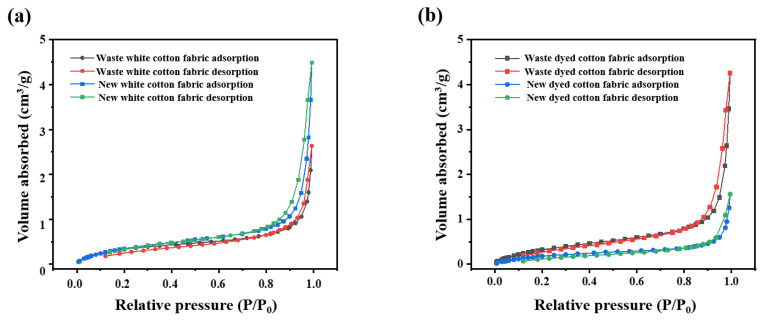
(**a**) Adsorption/desorption curves of used white fabric and new white fabric (**b**) Adsorption desorption curves of waste dyed fabric and new dyed fabric.

**Figure 8 polymers-17-00900-f008:**
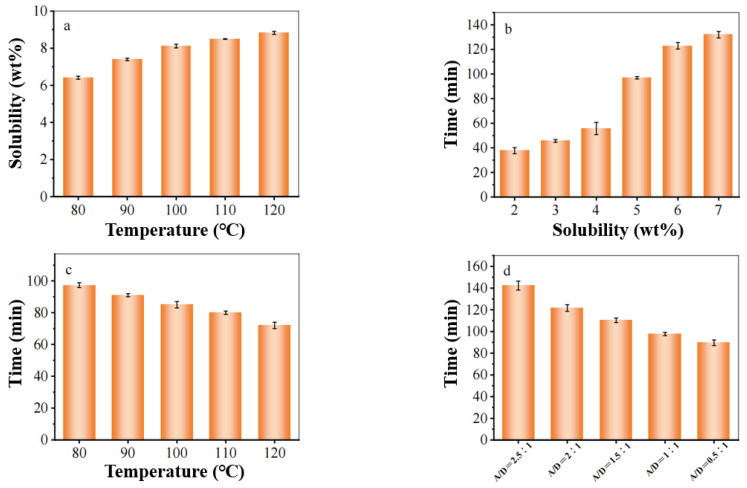
Dissolution temperature on solubility (**a**), waste cotton proportion on dissolution time (**b**), dissolution temperature on time (**c**), [AMIM]Cl concentration on time (**d**).

**Figure 9 polymers-17-00900-f009:**
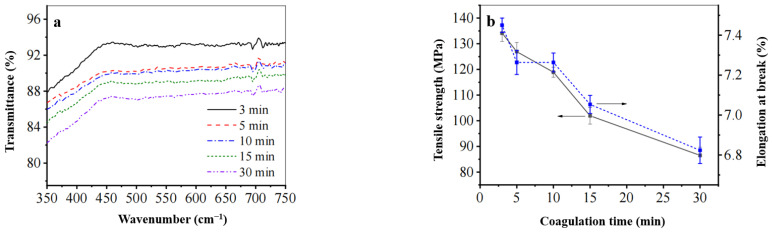
The effect of coagulation time on (**a**) the transmittance and (**b**) the mechanical properties of the regenerated cellulose film.

**Figure 10 polymers-17-00900-f010:**
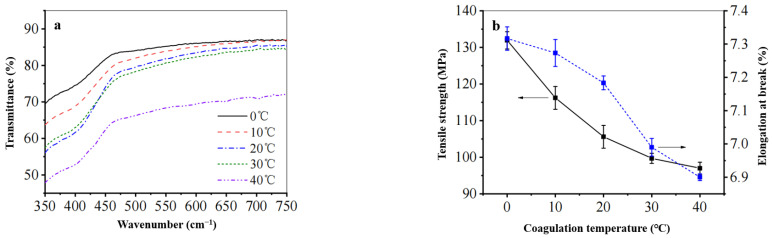
The effect of coagulation bath temperature on (**a**) the transmittance and (**b**) the mechanical properties of the regenerated cellulose film.

**Figure 11 polymers-17-00900-f011:**
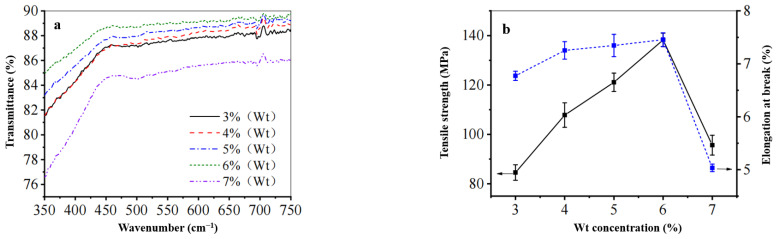
The effect of cellulose concentration on (**a**) the transmittance and (**b**) the mechanical properties of the regenerated cellulose film.

**Figure 12 polymers-17-00900-f012:**
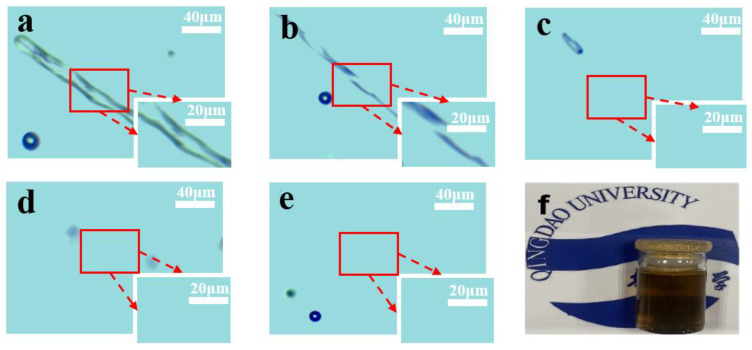
Microscopic images of waste cotton under different dissolution time ((**a**)—5 min; (**b**)—15 min; (**c**)—30 min; (**d**)—1 h; (**e**)—2 h) and (**f**) the solution photo after 1.5 h dissolution.

**Figure 13 polymers-17-00900-f013:**
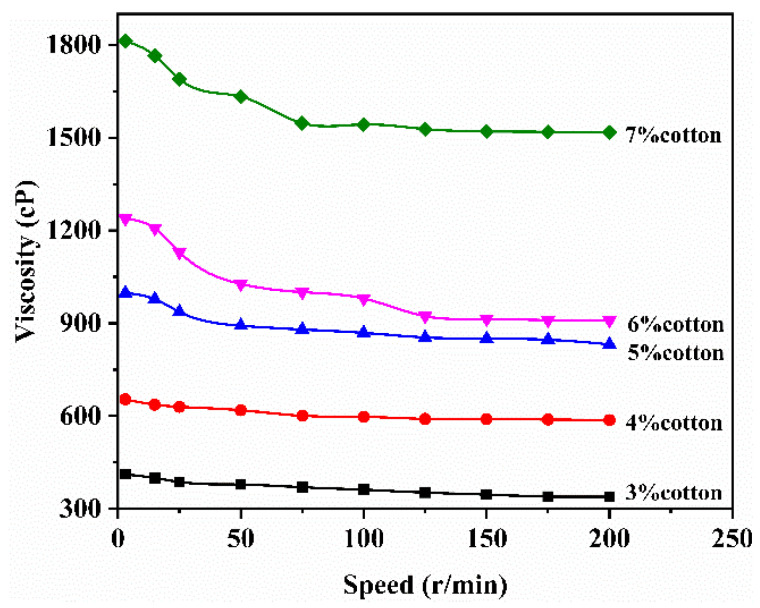
The curves of viscosity of waste cotton with different concentrations versus rotational speed.

**Figure 14 polymers-17-00900-f014:**
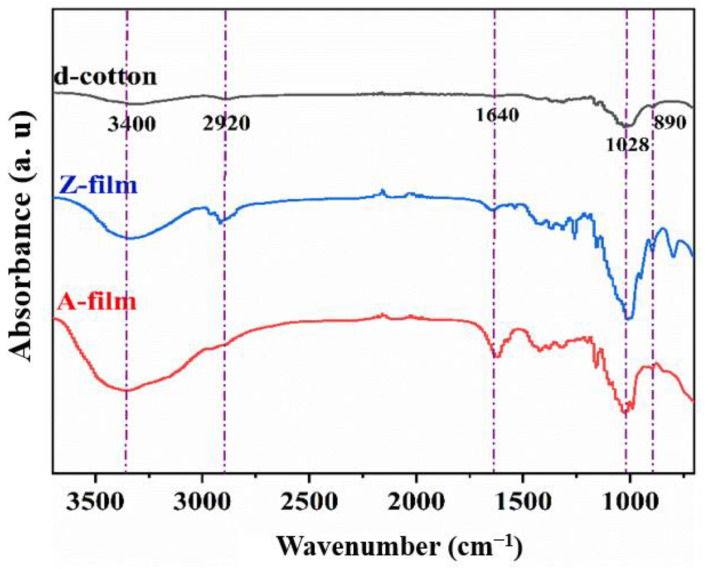
Infrared spectra of waste cotton after decolorization and the regenerated cellulose film.

**Figure 15 polymers-17-00900-f015:**
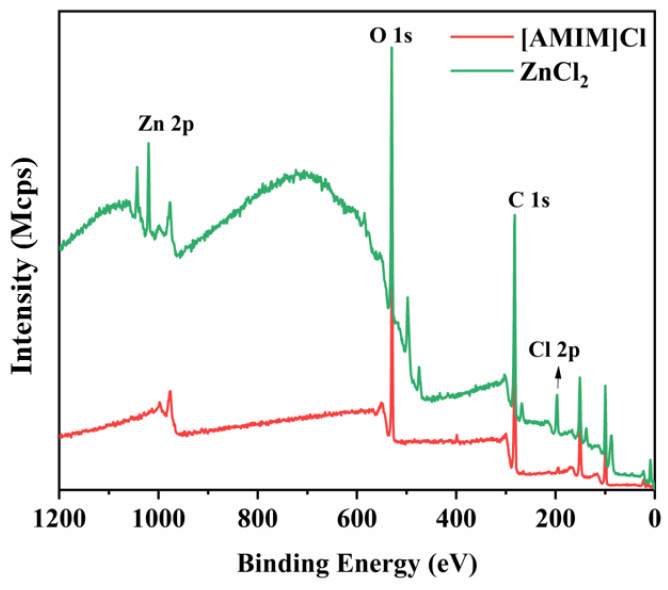
XPS spectra of the regenerated cellulose film.

**Figure 16 polymers-17-00900-f016:**
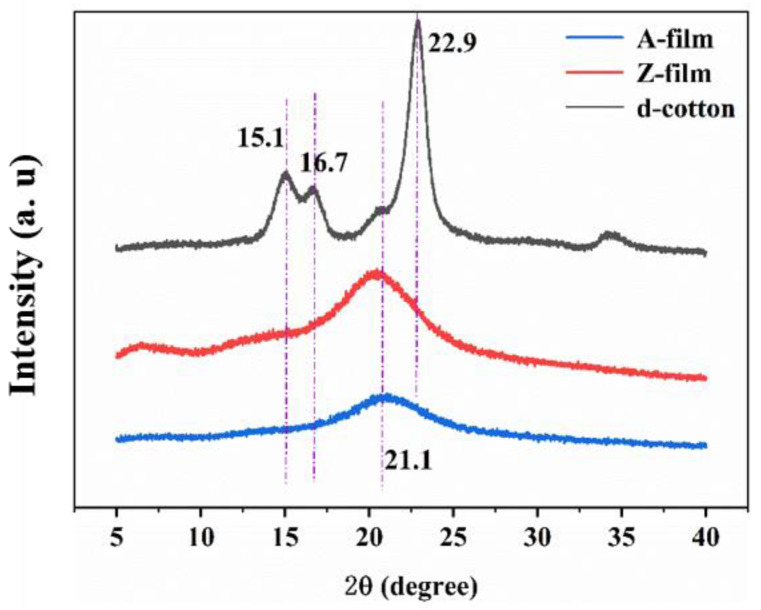
X-ray of waste cotton after decolorization and the regenerated cellulose film.

**Figure 17 polymers-17-00900-f017:**
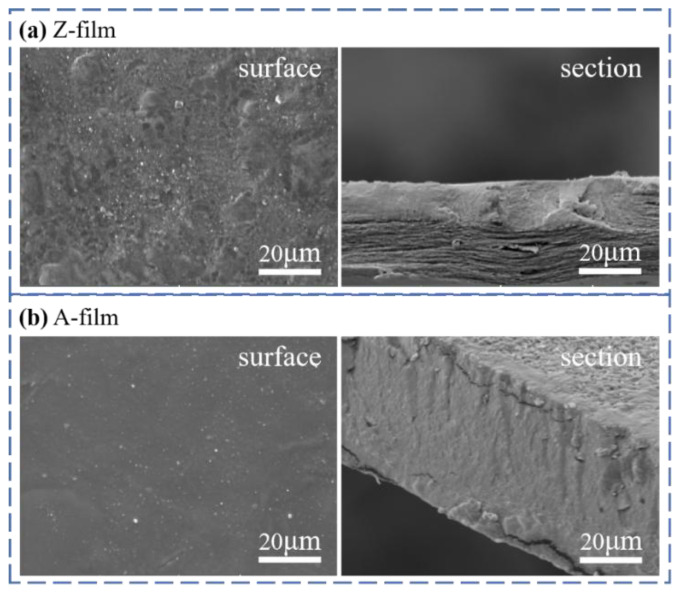
SEM images of the regenerated cellulose film.

## Data Availability

The original contributions presented in this study are included in this article, and further inquiries can be directed to the corresponding author.

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
