# Peer review of "Efficient Recovery of Waste Cotton Fabrics Using Ionic Liquid Methods"

_polymers, 2025, doi:10.3390/polym17070900_

Round 1

Reviewer 1 Report

Comments and Suggestions for Authors
  1. Add more relevant literature
  2. Authors have compared the both white and colored in new and waste cotton fabric form in terms of their chemical and performance characteristics, however I have sone doubt on the validity of this comparison. So my concern is that, have authors considered the cotton variety, yarn count, fabric construction parameters in new versus waste cotton fabric, as these parameters like cotton variet.yarn count have a huge effect on fabric properties like breaking strength.
  3. Authors should consider the above parameters while comparing the new and old fabric, and if is not possible then they should clearly mention in the manuscript.
  4. How was the decolorization process carried out, and does it have any impact on the dissolution and regeneration steps?
  5. Why authors chosen Ionic liquid 1-allyl-3-methylimidazole chloride 17 ([AMIM]Cl) and zinc chloride (ZnCl2) as solvents to dissolve decolorized waste cotton
  6. How authors determined the optimal dissolution conditions ?
  7. Have authors considered the environmental impact of the mentioned dissolution process?
  8. What are the potential applications of the regenerated cellulose films?
Comments on the Quality of English Language

language is fine.

Author Response

Comments 1: Add more relevant literature

Response 1: We appreciate your suggestions and will incorporate additional relevant documents into the manuscript.

Comments 2: Authors have compared the both white and colored in new and waste cotton fabric form in terms of their chemical and performance characteristics, however I have sone doubt on the validity of this comparison. So my concern is that, have authors considered the cotton variety, yarn count, fabric construction parameters in new versus waste cotton fabric, as these parameters like cotton variet. yarn count have a huge effect on fabric properties like breaking strength.

Response 2: In response to your questions, we have carefully considered the appropriate measures. As illustrated in Figure 1, we selected two identical pieces of white cotton fabric for our experiment: one that has been used daily and another that has been well-preserved and unused. For the colored fabric, although an exact match was not available, we chose a fabric of the same style but in a different color to maintain consistency.

Comments 3: Authors should consider the above parameters while comparing the new and old fabric, and if is not possible then they should clearly mention in the manuscript.

Response 3: Your suggestions are highly valuable, and we will clearly mark them in the manuscript. We have indeed strived to maintain consistency between old and new parameters, but this was not clearly expressed, which may have caused confusion for readers.

Comments 4: How was the decolorization process carried out, and does it have any impact on the dissolution and regeneration steps?

Response 4: The decolorization process employs the Na2S2O4-H2O2 method, with primary reagents being Na2S2O4 and H2O2, both of which cause minimal damage to the fabric. SEM observations reveal that the fabric's surface remains largely unchanged before and after decolorization. The crystallinity of the fabric increases from 62.8% (raw cotton) to 65.7%, while the degree of polymerization decreases from approximately 1327 to about 735, facilitating subsequent dissolution and regeneration of the fabric.

Comments 5: Why authors chosen Ionic liquid 1-allyl-3-methylimidazole chloride 17 ([AMIM]Cl) and zinc chloride (ZnCl2) as solvents to dissolve decolorized waste cotton

Response 5:  Ionic liquid 1-allyl-3-methylimidazole chloride ([AMIM]Cl) is often referred to as the "solvent of the future" due to its low vapor pressure, non-volatility, ease of recovery, and lack of harmful gas emissions during industrial use. It also exhibits superior thermal stability and electrical conductivity compared to traditional solvents. Aqueous ZnCl2, first reported as a cellulose solvent in 1932, is an environmentally friendly metal inorganic salt with negligible environmental toxicity or volatility. Importantly, these salts are easily recyclable.

Comments 6: How authors determined the optimal dissolution conditions?

Response 6: As shown in Figure 8, we investigated the impact of dissolution temperature on the solubility and dissolution time of waste cotton. Increasing temperature enhances solubility and reduces dissolution time. However, higher temperatures complicate stirring, so we selected a 6% concentration for the film-making solution. Adding DMSO reduces the viscosity of the film-making solution. Therefore, the optimal dissolution conditions are as follows: [AMIM]Cl: DMSO = 1:1, dissolution temperature of 110°C, dissolution time of 120 minutes, and a 6% concentration of the film-making solution.

Comments 7: Have authors considered the environmental impact of the mentioned dissolution process?

Response 7: During the dissolution process, no harmful gases are produced, and the ionic liquid exhibits excellent thermal stability. The process operates at low temperatures without requiring pressure, minimizing environmental impact and reducing energy consumption during production.

Comments 8: What are the potential applications of the regenerated cellulose films?

Response 8: Regenerated cellulose films find extensive applications in packaging, medical treatment, agriculture, and environmental protection. In environmental protection, recycled cellulose films can substitute for degradable plastic products such as garbage bags and disposable tableware, helping to reduce white pollution. In agriculture, regenerated cellulose films can be used for agricultural mulch and seedling pots, providing effective thermal insulation and moisture retention. In packaging, recycled cellulose films can be utilized in food and pharmaceutical packaging, enhancing product shelf life and safety.

Reviewer 2 Report

Comments and Suggestions for Authors

This paper evaluated the effect of [AMIM]Cl and ZnCl2 on the dissolution and decolorisation of dyed textile waste.

The study was original but the scientific merit is not high. The study was straightforward and it only provided little new information to this area.

1. Authors should analyse more different types of dyed materials. One major concern is the lack of statistical analysis.

2. The results are nicely presented, but there was no statistical analysis.

3. About the tables and figures,  there is no statistical analysis of all data.

Author Response

Comments 1: Authors should analyse more different types of dyed materials. One major concern is the lack of statistical analysis

Response 1: We have done corresponding analysis of different types of dyeing materials, and more different types of dyeing materials will be presented in future papers of our research group. We will add the corresponding statistical analysis to the data in this article.

Comments 2: The results are nicely presented, but there was no statistical analysis

Response 2: Thanks for your valuable comments, statistical analysis of the resulting data has been added to the conclusion

Comments 3: About the tables and figures, there is no statistical analysis of all data.

Response 3: Thank you for your comments and the corresponding data has been analyzed.

Reviewer 3 Report

Comments and Suggestions for Authors

1- It is suggested that lines 2 and 3 of the abstract be deleted. The most important achievement of the paper should be added to this section.

2- The main innovation of this research is not clear. This problem should be resolved at the end of the introduction. In addition, the engineering approach of this research should be clarified.

3- Add the year corresponding to the standards used (ASTM).

4- Do changes in environmental conditions or pH affect the final result obtained?

5- Please explain the time of use or environmental conditions during the use of the waste fabrics used and the effect of time and conditions of use on the results obtained.

6- Why are the end points in the IBT test different for waste and new fabrics? Does the type of dyeing and the dyestuff used affect the results obtained in this test?

7- Why did the authors not use the Raman test to examine the functional groups and used FTIR? Also, how was the sample preparation done in the FTIR test?

Author Response

Comments 1: It is suggested that lines 2 and 3 of the abstract be deleted. The most important achievement of the paper should be added to this section
Response 1: We appreciate your valuable suggestions and have revised the paper accordingly.

Comments 2: The main innovation of this research is not clear. This problem should be resolved at the end of the introduction. In addition, the engineering approach of this research should be clarified
Response 2: We thank you for your suggestion. This research aims to address the challenges of large-scale waste cotton fabric generation and recycling difficulties. Firstly, we compared the performance of old and new cotton fabrics, followed by regenerating waste cotton fabric using ionic liquids. Your aforementioned suggestions have been incorporated into the article.

Comments 3: Add the year corresponding to the standards used (ASTM)
Response 3: There is no corresponding ASTM standard for this; however, an applicable ISO standard (ISO 5351:2010) has been referenced in the text.

Comments 4: Do changes in environmental conditions or pH affect the final result obtained?
Response 4: Environmental conditions can influence various properties of the fabric. To ensure consistency across variables, both white and colored waste fabrics were sourced from a single provider.

Comments 5: Please explain the time of use or environmental conditions during the use of the waste fabrics used and the effect of time and conditions of use on the results obtained.
Response 5: Waste fabric samples were provided by experimental personnel after more than one month of wear in a coastal summer environment with daily humidity exceeding 70% and temperatures around 30°C. Prolonged wear time affects fabric properties due to sweat and oil secretions, leading to decreased strength and polymerization.

Comments 6: Why are the end points in the IBT test different for waste and new fabrics? Does the type of dyeing and the dyestuff used affect the results obtained in this test?
Response 6: To our knowledge, the type of dye used does not impact test results as dye molecules are small and attach to the fiber surface via hydrogen bonding.

Comments 7: Why did the authors not use the Raman test to examine the functional groups and used FTIR? Also, how was the sample preparation done in the FTIR test?
Response 7: Cellulose is an organic compound, and FTIR is more advantageous for identifying organic compounds compared to Raman spectroscopy. Many cotton fabrics undergo fluorescent brightening treatment at the factory, which can significantly affect Raman tests. In the FTIR test, the sample is first ground into powder using a mortar, then prepared as a tablet with KBr, and finally tested by transmission.

Round 2

Reviewer 1 Report

Comments and Suggestions for Authors

Thank you for your response to the reviewers' comments. However, upon reviewing your revised manuscript and response letter, we have identified a significant concern regarding the incorporation of feedback related to crucial experimental parameters.

  • For example, regarding comment #2 and 3,” have authors considered the cotton variety, yarn count, and fabric construction parameters in new versus waste cotton fabric, as these parameters like cotton variety. yarn count have a huge effect on fabric properties like breaking strength.”

Please clearly state in the manuscript:

  1. Whether these parameters were controlled/considered.
  2. If yes, provide details in Materials and Methods.
  3. If no, discuss limitations in the discussion.

Provide a detailed explanation of these changes in your revised response file.

  • Comment #5: Why have authors chosen Ionic liquid 1-allyl-3-methylimidazole chloride 17 ([AMIM]Cl) and zinc chloride (ZnCl2) as solvents to dissolve decolorized waste cotton”,

Add literature to the manuscript to justify your response on this comment.

  • Comments 7: Have authors considered the environmental impact of the mentioned dissolution process? Response 7: During the dissolution process, no harmful gases are produced, and the ionic liquid exhibits excellent thermal stability. The process operates at low temperatures without requiring pressure, minimizing environmental impact and reducing energy consumption during production.

Add this response in the discussion section of the manuscript with proper referencing from literature where required.

Author Response

Comments 1:

For example, regarding comment #2 and 3,” have authors considered the cotton variety, yarn count, and fabric construction parameters in new versus waste cotton fabric, as these parameters like cotton variety. yarn count have a huge effect on fabric properties like breaking strength.”

Please clearly state in the manuscript:

  1. Whether these parameters were controlled/considered.
  2. If yes, provide details in Materials and Methods.
  3. If no, discuss limitations in the discussion.

Provide a detailed explanation of these changes in your revised response file.

Response 1:

Thank you for your suggestion, and the corresponding statement has been added to 3.1.1, the analysis of breaking strength.

The breaking strength of fabrics is significantly influenced by parameters such as cotton variety, yarn count, and fabric structure. In this study, all fabrics were sourced from online platforms, resulting in a considerable degree of randomness that makes it challenging to control these parameters. Among these factors, the cotton variety can be identified; specifically, the fabric used is consistent with fine cashmere cotton. Furthermore, all fabrics employed in this experiment are thin woven types, exhibiting minimal variation in structural parameters.

It is important to note that the yarn count for each fabric remains unspecified. A higher yarn count indicates greater linear density and enhanced breaking strength. Consequently, the breaking strength data presented in this study serves only as a preliminary comparison between new and old cotton varieties. For more detailed analysis, it would be necessary to further reduce variables and conduct more rigorous experimental research.

Comments 2:

Comment #5: Why have authors chosen Ionic liquid 1-allyl-3-methylimidazole chloride 17 ([AMIM]Cl) and zinc chloride (ZnCl2) as solvents to dissolve decolorized waste cotton”,

Add literature to the manuscript to justify your response on this comment.

Response 2:

 The corresponding references have been added in the introduction section of the article

Comments 3:

Comments 7: Have authors considered the environmental impact of the mentioned dissolution process? Response 7: During the dissolution process, no harmful gases are produced, and the ionic liquid exhibits excellent thermal stability. The process operates at low temperatures without requiring pressure, minimizing environmental impact and reducing energy consumption during production.

Add this response in the discussion section of the manuscript with proper referencing from literature where required.

Response 3:

The corresponding responses and references have been added to article 3.2.1.

Reviewer 2 Report

Comments and Suggestions for Authors

I am fine with the revisions that provided additional information.

Comments on the Quality of English Language

Fine.

Author Response

Thanks to your suggestions, we have further refined some of the languages.

Round 3

Reviewer 1 Report

Comments and Suggestions for Authors

Thank you for incorporating the suggested revisions. 

Author Response

Thank you for your advise!